# Bayesian Bellman Operators

**Matthew Fellows**[*]   **Kristian Hartikainen**   **Shimon Whiteson**
Department of Computer Science
University of Oxford

## Abstract

We introduce a novel perspective on Bayesian reinforcement learning (RL); whereas existing approaches infer a posterior over the transition distribution or $Q$-function, we characterise the uncertainty in the Bellman operator. Our Bayesian Bellman operator (BBO) framework is motivated by the insight that when bootstrapping is introduced, model-free approaches actually infer a posterior over Bellman operators, not value functions. In this paper, we use BBO to provide a rigorous theoretical analysis of model-free Bayesian RL to better understand its relationship to established frequentist RL methodologies. We prove that Bayesian solutions are consistent with frequentist RL solutions, even when approximate inference is used, and derive conditions for which convergence properties hold. Empirically, we demonstrate that algorithms derived from the BBO framework have sophisticated deep exploration properties that enable them to solve continuous control tasks at which state-of-the-art regularised actor-critic algorithms fail catastrophically.

## 1   Introduction

A Bayesian approach to reinforcement learning (RL) characterises uncertainty in the Markov decision process (MDP) via a posterior [35, 78]. A great advantage of Bayesian RL is that it offers a natural and elegant solution to the exploration/exploitation problem, allowing the agent to explore to reduce uncertainty in the MDP, but only to the extent that exploratory actions lead to greater expected return; unlike in heuristic strategies such as $\varepsilon$-greedy and Boltzmann sampling, the agent does not waste samples trying actions that it has already established are suboptimal, leading to greater sampling efficiency. Elementary decision theory shows that the only admissible decision rules are Bayesian [22] because a non-Bayesian decision can always be improved upon by a Bayesian agent [24]. In addition, pre-existing domain knowledge can be formally incorporated by specifying priors.

In model-free Bayesian RL, a posterior is inferred over the $Q$-function by treating samples from the MDP as stationary labels for Bayesian regression. A major theoretical issue with existing model-free Bayesian RL approaches is their reliance on bootstrapping using a $Q$-function approximator, as samples from the exact $Q$-function are impractical to obtain. This introduces error as the samples are no long estimates of a $Q$-function and their dependence on the approximation is not accounted for. It is unclear what posterior, if any, these methods are inferring and how it relates to the RL problem.

In this paper, we introduce Bayesian Bellman Operators (BBO), a novel model-free Bayesian RL framework that addresses this issue and facilitates a theoretical exposition of the relationship between model-free Bayesian and frequentist RL approaches. Using our framework, we demonstrate that, by bootstrapping, model-free Bayesian RL infers a posterior over *Bellman operators*. For our main contribution, we prove that the BBO posterior concentrates on the true Bellman operator (or the closest representation in our function space of Bellman operators). Hence a Bayesian method using the BBO posterior is consistent with the equivalent frequentist solution in the true MDP. We derive convergent gradient-based approaches for Bayesian policy evaluation and uncertainty estimation. Remarkably, our consistency and convergence results still hold when approximate inference is used.

---

[*]Correspondence to `matthew.fellows@cs.ox.ac.uk`

35th Conference on Neural Information Processing Systems (NeurIPS 2021).

Our framework is general and can recover empirically successful algorithms such as BootDQNprior+ [57]. We demonstrate that BootDQNprior+'s lagged target parameters, which are essential to its performance, arise from applying approximate inference to the BBO posterior. Lagged target parameters cannot be explained by existing model-free Bayesian RL theory. Using BBO, we extend BootDQNprior+ to continuous domains by developing an equivalent Bayesian actor-critic algorithm. Our algorithm can learn optimal policies in domains where state-of-the-art actor-critic algorithms like soft actor-critic [39] fail catastrophically due to their inability to properly explore.

## 2 Bayesian Reinforcement Learning

*To aid the reader with notation, we provide a mathematical glossary in Appendix A.*

### 2.1 Preliminaries

Formally, an RL problem is modelled as a Markov decision process (MDP) defined by the tuple $\langle \mathcal{S}, \mathcal{A}, r, P, P_0, \gamma \rangle$ [72, 60], where $\mathcal{S}$ is the set of states and $\mathcal{A}$ the set of available actions. At time $t$, an agent in state $s_t \in \mathcal{S}$ chooses an action $a_t \in \mathcal{A}$ according to the policy $a_t \sim \pi(\cdot|s_t)$. The agent transitions to a new state according to the state transition distribution $s_{t+1} \sim P(\cdot|s_t, a_t)$ which induces a scalar reward $r_t := r(s_{t+1}, a_t, s_t) \in \mathbb{R}$ with $\sup_{s', a, s} |r(s', a, s)| < \infty$. The initial state distribution for the agent is $s_0 \sim P_0$ and the state-action transition distribution is denoted as $P^\pi(s', a'|s, a)$, which satisfies $dP^\pi(s', a'|s, a) = d\pi(a'|s')dP(s'|s, a)$.. As the agent interacts with the environment it gathers a trajectory: $(s_0, a_0, r_0, s_1, a_1, r_1, s_2...)$. We seek an optimal policy $\pi^* \in \arg\max_\pi J^\pi$ that maximises the total expected discounted return: $J^\pi := \mathbb{E}_\pi \left[ \sum_{t=0}^\infty \gamma^t r_t \right]$ where $\mathbb{E}_\pi$ is the expectation over trajectories induced by $\pi$. The $Q$-function is the total expected reward as a function of a state-action pair: $Q^\pi(s, a) := \mathbb{E}_\pi[\sum_{t=0}^\infty r_t|s_0 = s, a_0 = a]$. Any $Q$-function satisfies the Bellman equation $\mathcal{B}[Q^\pi] = Q^\pi$ where the Bellman operator is defined as:

$$\mathcal{B}[Q^\pi](s, a) := \mathbb{E}_{P^\pi(s', a'|s, a)} \left[ r(s', a, s) + \gamma Q^\pi(s', a') \right]. \tag{1}$$

### 2.2 Model-based vs Model-free Bayesian RL

Bayes-adaptive MDPs (BAMDPs) [27] are a framework for *model-based* Bayesian reinforcement learning where a posterior marginalises over the uncertainty in the unknown transition distribution and reward functions to derive a Bayesian MDP. BAMDP optimal policies are the gold standard, optimally balancing exploration with exploitation but require learning a model of the unknown transition distribution which is typically challenging due to its high-dimensionality and multi-modality [67]. Furthermore, planning in BAMDPs requires the calculation of high-dimensional integrals which render the problem intractable. Even with approximation, most existing methods are restricted to small and discrete state-action spaces [6, 38]. One notable exception is VariBAD [82] which exploits a meta learning setting to carry out approximate Bayesian inference. Unfortunately this approximation sacrifices the BAMDP's theoretical properties and there are no convergence guarantees.

Existing model-free Bayesian RL approaches attempt to solve a Bayesian regression problem to infer a posterior predictive over a value function [78, 35]. Whilst foregoing the ability to separately model reward uncertainty and transition dynamics, modelling uncertainty in a value function avoids the difficulty of estimating high dimensional conditional distributions and mimics a Bayesian regression problem, for which there are tractable approximate methods [44, 10, 47, 61, 33, 51]. These methods assume access to a dataset of $N$ samples: $\mathcal{D}^N := \{q_i\}_{i=1:N}$ from a distribution over the true $Q$-function at each state-action pair: $q_i \sim P_Q(\cdot|s_i, a_i)$. Each sample is an estimate of a point of the true $Q$-function $q_i = Q^\pi(s_i, a_i) + \eta_i$ corrupted by noise $\eta_i$. By introducing a probabilistic model of this random process, the posterior over the $Q$-function $P(Q^\pi|s, a, \mathcal{D}^N)$ can be inferred, which characterises the aleatoric uncertainty in the sample noise and epistemic uncertainty in the model. Modeling aleatoric uncertainty is the goal of distributional RL [11]. In Bayesian RL we are more concerned with epistemic uncertainty, which can be reduced by exploration [57].

### 2.3 Theoretical Issues with Existing Approaches

Unfortunately for most settings it is impractical to sample directly from the true $Q$-function. To obtain efficient algorithms the samples $q_i$ are approximated using bootstrapping: here a parametric function approximator $\hat{Q}_\omega : \mathcal{S} \times \mathcal{A} \to \mathbb{R}$ parametrised by $\omega \in \Omega$ is learnt as an approximation of the $Q$-function $\hat{Q}_\omega \approx Q^\pi$ and then a TD sample is used in place of $q_i$. For example a one-step TD estimate approximates the samples as: $q_i \approx r_i + \gamma \hat{Q}_\omega(s_i, a_i)$, introducing an error that is dependent on $\omega$. Existing approaches do not account for this error's dependency on the function approximator.

Samples are no longer noisy estimates of a point $Q^\pi(s_i, a_i)$ and the resulting posterior predictive is not $P(Q^\pi|s, a, \mathcal{D}^N)$ as it has dependence on $\hat{Q}_\omega$ due to the dataset. This problem is made worse when a posterior is inferred over an optimal $Q$-function as it is impossible to sample from the optimal policy a priori to obtain unbiased samples. This is a major theoretical issue that raises the following questions:

1. Do model-free Bayesian RL approaches that use bootstrapping still infer a posterior?

2. If it exists, how does this posterior relate to solving the RL problem?

3. What effect does approximate inference have on the solution?

4. Do methods that sample from an approximate posterior converge?

**Contribution:** Our primary contribution is to address these questions by introducing the BBO framework. In answer to Question 1, BBO shows that, by introducing bootstrapping, we actually infer a posterior over Bellman operators. We can use this posterior to marginalise over all Bellman operators to obtain a Bayesian Bellman operator. Our theoretical results provide answers to Questions 2-4, proving that the Bayesian Bellman operator can parametrise a TD fixed point as the number of samples $N \to \infty$ and is analogous to the projection operator used in convergent reinforcement learning. Our results hold even under posterior approximation. Although our contributions are primarily theoretical, many of the benefits afforded by Bayesian methods play a significant role in a wide range of real-world applications of RL where identifying decisions that are being made under high uncertainty is crucial. We discuss the impact of our work further in Appendix B.

## 3 Bayesian Bellman Operators

*Detailed proofs and a discussion of assumptions for all theoretical results are found in Appendix C.*

To introduce the BBO framework we consider the Bellman equation using a function approximator: $\mathcal{B}[\hat{Q}_\omega] = \hat{Q}_\omega$. Using Eq. (1), we can write the Bellman operator for $\hat{Q}_\omega$ as an expectation of the empirical Bellman function $b_\omega$:

$$\mathcal{B}[\hat{Q}_\omega](s,a) = \mathbb{E}_{P^\pi(s',a'|a,s)}\left[b_\omega(s',a',s,a)\right], \quad b_\omega(s',a',s,a) := r(s',a,s) + \gamma\hat{Q}_\omega(s',a'). \quad (2)$$

When evaluating the Bellman operator to solve the Bellman equation, we can evaluate the function approximator $\hat{Q}_\omega(s,a)$ but we cannot evaluate $\mathcal{B}[\hat{Q}_\omega](s,a)$ due to the uncertainty in reward function and transition distribution. In BBO we capture this uncertainty by treating the empirical Bellman function as a transformation of variables $b_\omega(\cdot, s, a) : \mathcal{S} \times \mathcal{A} \to \mathbb{R}$ for each $(s, a)$. The transformed variable $B : \mathbb{R} \to \mathbb{R}$ has a conditional distribution $P_B(b|s, a, \omega)$ which is the *pushforward* of $P^\pi(s', a'|s, a)$ under the transformation $b_\omega(\cdot, s, a)$. For any $P_B$-integrable function $f : \mathbb{R} \to \mathbb{R}$, the pushforward satisfies:

$$\mathbb{E}_{P_B(b|s,a,\omega)}\left[f(b)\right] = \mathbb{E}_{P^\pi(s',a'|s,a)}\left[f \circ b_\omega(s',a',s,a)\right]. \quad (3)$$

As the pushforward $P_B(b|s, a, \omega)$ is a distribution over empirical Bellman functions, each sample $b \sim P_B(\cdot|s, a, \omega)$ is a noisy sample of the Bellman operator at a point: $b_i = \mathcal{B}[\hat{Q}_\omega](s_i, a_i) + \eta_i$. To prove this, observe that taking expectations of $b$ recovers $\mathcal{B}[\hat{Q}_\omega](s, a)$:

$$\mathbb{E}_{P_B(b|s,a,\omega)}[b] \underbrace{=}_{\text{Eq. (3)}} \mathbb{E}_{P^\pi(s',a'|s,a)}\left[b_\omega(s',a',s,a)\right] \underbrace{=}_{\text{Eq. (2)}} \mathcal{B}[\hat{Q}_\omega](s,a).$$

As the agent interacts with the environment, it obtains samples from the transition distribution $s_i' \sim P(\cdot|s_i, a_i)$ and policy $a_i' \sim \pi(\cdot|s_i')$. From Eq. (3) a sample from the distribution $b_i \sim P_B(\cdot|s_i, a_i, \omega)$ is obtained from these state-action pairs by applying the empirical Bellman function $b_i = r_i + \gamma\hat{Q}_\omega(s_i', a_i')$. As we discussed in Section 2.3, existing model-free Bayesian RL approaches incorrectly treat each $b_i$ as a sample from a distribution over the value function $P_Q(Q^\pi|s, a)$. BBO corrects this by modelling the true conditional distribution: $P_B(b|s, a, \omega)$ that generates the data.

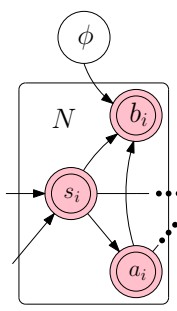

The graphical model for BBO is shown in Fig. 1. To model $P_B(b|s, a, \omega)$ we assume a parametric conditional distribution: $P(b|s, a, \phi)$ with model parameters

Figure 1: Graphical Model for BBO.

$\phi \in \Phi$ and a conditional mean: $\mathbb{E}_{P(b|s,a,\phi)}[b] = \hat{B}_\phi(s, a)$. It is also possible to specify a nonparametric model: $P(b|s, a)$. The conditional mean of the distribution $\hat{B}_\phi$ defines a function space of approximators that represents a space of Bellman operators, each indexed by $\phi \in \Phi$. The choice of $P(b|s, a, \phi)$ should therefore ensure that the space of approximate Bellman operators characterised by $\hat{B}_\phi$ is expressive enough to sufficiently represent the true Bellman operator. As we are not concerned with modelling the transition distribution in our model-free paradigm, we assume states are sampled either from an ergodic Markov chain, or i.i.d. from a buffer. Off-policy samples can be corrected using importance sampling.

**Assumption 1** (State Generating Distribution). *Each state $s_i$ is drawn either i) i.i.d. from a distribution $\rho(s)$ with support over $S$ or ii) from an ergodic Markov chain with stationary distribution $\rho(s)$ defined over a $\sigma$-algebra that is countably generated from $S$.*

We represent our preexisting beliefs in the true Bellman operator by specifying a prior $P(\phi)$ with a density $p(\phi)$ which assigns mass over parameterisations of function approximators $\phi \in \Phi$ in accordance with how well we believe they represent $\mathcal{B}[\hat{Q}_\omega]$. Given the prior and a dataset $\mathcal{D}_\omega^N :=$ $\{b_i, s_i, a_i\}_{i=1:N}$ of samples from the true distribution $P_B$, we infer the posterior $P(\phi|\mathcal{D}_\omega^N)$ using Bayes' rule which has the density (see Appendix D.1 for a derivation using both state generating distributions of Assumption 1):

$$p(\phi|\mathcal{D}_\omega^N) = \frac{\prod_{i=1}^N p(b_i|s_i, a_i, \phi)p(\phi)}{\int_\Phi \prod_{i=1}^N p(b_i|s_i, a_i, \phi)dP(\phi)}. \tag{4}$$

To be able to make predictions, we infer the posterior predictive: $p(b|\mathcal{D}_\omega^N, s, a) :=$ $\int_\Phi p(b|s, a, \phi)dP(\phi|\mathcal{D}_\omega^N)$. Unlike existing approaches, our posterior density is a function of $\omega$, which correctly accounts for the dependence on $\hat{Q}_\omega$ in our data and the generating distribution $P_B(b|s, a, \omega)$. We highlight that it is possible to define a likelihood and prior that are functions of $\omega$ to encode any prior knowledge of how the underlying Bellman operator varies with $\omega$, however this is not strictly necessary as the posterior automatically accounts for this dependence due to its conditioning on $\mathcal{D}_\omega^N$. As we anticipate that most applications of BBO will seek to learn an optimal policy, and hence and optimal $Q$-function, a prior that incorporates any knowledge available about the optimal Bellman operator will speed learning and give the agent an advantage.

As our data depends on $\hat{Q}_\omega$, we must introduce a method of learning the correct $Q$-function approximator. As every Bellman operator characterises an MDP, the posterior predictive mean represents a Bayesian estimate of the true MDP by using the posterior to marginalise over all Bellman operators that our model can represent according to our uncertainty in their value:

$$\mathcal{B}_{\omega,N}^\star(s, a) := \mathbb{E}_{P(b|\mathcal{D}_\omega^N, s, a)}[b] = \mathbb{E}_{P(\phi|\mathcal{D}_\omega^N)}\left[\hat{B}_\phi(s, a)\right]. \tag{5}$$

For this reason, we refer to the predictive mean $\mathcal{B}_{\omega,N}^\star$ as the *Bayesian Bellman operator* and our $Q$-function approximator should satisfy a Bellman equation using $\mathcal{B}_{\omega,N}^\star$. Our objective is therefore to find $\omega^\star$ such that $\hat{Q}_{\omega^\star} = \mathcal{B}_{\omega^\star,N}^\star$. A simple approach to learn $\omega^\star$ is to minimise the mean squared Bayesian Bellman error (MSBBE) between the posterior predictive and function approximator:

$$\text{MSBBE}_N(\omega) := \left\|\hat{Q}_\omega - \mathcal{B}_{\omega,N}^\star\right\|_{\rho,\pi}^2 \tag{6}$$

Here the distribution on the $\ell_2$-norm is $\rho(s)\pi(a|s)$ where recall $\rho(s)$ is defined in Assumption 1. Although the MSBBE has a similar form to a mean squared Bellman error with a Bayesian Bellman operator in place of the Bellman operator, our theoretical results in Section 3.1 show its frequentist interpretation is closer to the mean squared projected Bellman operator used by convergent TD algorithms [70]. We derive the MSBBE gradient in Appendix D.3:

$$\nabla_\omega \text{MSBBE}_N(\omega)$$
$$= \mathbb{E}_{\rho,\pi}\left[\left(\hat{Q}_\omega - \mathbb{E}_{P(\phi|\mathcal{D}_\omega^N)}\left[\hat{B}_\phi\right]\right)\left(\nabla_\omega \hat{Q}_\omega - \mathbb{E}_{P(\phi|\mathcal{D}_\omega^N)}\left[\hat{B}_\phi \nabla_\omega \log p(\phi|\mathcal{D}_\omega^N)\right]\right)\right]. \tag{7}$$

If we can sample from the posterior then unbiased estimates of $\nabla_\omega \text{MSBBE}_N(\omega)$ can be obtained, hence minimising the MSBBE via a stochastic gradient descent algorithm is convergent if the standard Robbins-Munro conditions are satisfied [62]. When existing approaches are used, the posterior has

no dependence on $\omega$ and the gradient $\nabla_\omega \log p(\phi|\mathcal{D}_\omega^N)$ is not accounted for, leading to gradient terms being dropped in the update. Stochastic gradient descent using these updates does not optimise any objective and so may not converge to any solution. The focus of our analysis in Section 4.1 is to extend convergent gradient methods for minimising the MSSBE to approximate inference techniques in situations where sampling from the posterior becomes intractable.

Minimising the MSBBE also avoids the double sampling problem encountered in frequentist RL where to minimise the mean squared Bellman error, two independent samples from $P(s'|s,a)$ are required to obtain unbiased gradient estimates [7]. In BBO, this issue is avoided by drawing two independent approximate Bellman operators $B_{\phi_1}$ and $B_{\phi_2}$ from the posterior $\phi_1, \phi_2 \sim P(\cdot|\mathcal{D}_\omega^N)$ instead.

### 3.1 Consistency of the Posterior

To address Question 2, we develop a set of theoretical results to understand the posterior's relationship to the RL problem. We introduce some mild regularity assumptions on our choice of model:

**Assumption 2** (Regularity of Model). *i)* $\hat{Q}_\omega$ *is bounded and* $(\Phi, d_\Phi)$ *and* $(\Omega, d_\Omega)$ *are compact metric spaces; ii)* $\hat{B}_\phi$ *is Lipschitz in* $\phi$, $P(b|s,a,\phi)$ *has finite variance and a density* $p(b|s,a,\phi)$ *which is Lipschitz in* $\phi$ *and bounded; and iii)* $p(\phi) \propto \exp\left(-R(\phi)\right)$ *where* $R(\phi)$ *is bounded and Lipschitz.*

Our main result is a Bernstein-von-Mises-type theorem [49] applied to reinforcement learning. We prove that the posterior asymptotically converges to a Dirac delta distribution centered on the set of parameters that minimise the KL divergence between the true and model distributions, which are the frequentist maximum likelihood parameters:

$$\phi_\omega^\star := \underset{\phi \in \Phi}{\arg\min} \, \text{KL}(P_B(b,s,a|\omega) \parallel P(b,s,a|\phi)) = \underset{\phi \in \Phi}{\arg\min} \, \mathbb{E}_{P_B(b,s,a|\omega)}\left[-\log p(b,s,a|\phi)\right], \quad (8)$$

where the expectation is taken with respect to distribution that generates the data: $P_B(b,s,a|\omega)$ that satisfies $dP_B(b,s,a|\omega) = dP_B(b|s,a,\omega)d\pi(a|s)d\rho(s)$. We make a simplifying assumption that there is a single maximum likelihood parameter, which eases analysis and exposition of our results. We discuss the more general case where it does not hold in Appendix C.3.

**Assumption 3** (Single Minimiser). *The set of maximum likelihood parameters* $\phi_\omega^\star$ *defined in Eq.* (8) *exists and is a singleton.*

**Theorem 1.** *Under Assumptions 1-3 in the limit* $N \to \infty$ *the posterior concentrates weakly on* $\phi_\omega^\star$: *i)* $P(\phi|\mathcal{D}_\omega^N) \Longrightarrow \delta(\phi = \phi_\omega^\star)$ *a.s.; ii)* $\mathcal{B}_{\omega,N}^\star \xrightarrow{a.s.} \hat{B}_{\phi_\omega^\star}$; *and iii)* $MSBBE_N(\omega) \xrightarrow{a.s.} \|\hat{Q}_\omega - \hat{B}_{\phi_\omega^\star}\|_{\rho,\pi}^2$.

If our model can sufficiently represent the true conditional distribution then $\text{KL}(P_B(b,s,a|\omega) \parallel P(b,s,a|\phi_\omega^\star)) = 0 \implies P_B(b|s,a,\omega) = P(b|s,a,\phi_\omega^\star)$. Theorem 1 proves that the posterior concentrates on the frequentist solution $\phi_\omega^\star$ and hence the Bayesian Bellman operator converges to the true Bellman operator: $\hat{B}_{\phi_\omega^\star}(s,a) = \mathbb{E}_{P(b|s,a,\phi_\omega^\star)}[b] = \mathbb{E}_{P_B(b|s,a,\omega)}[b] = \mathcal{B}[\hat{Q}_\omega](s,a)$. As every Bellman operator characterises an MDP, any Bayesian RL solution obtained using the BBO posterior such as an optimal policy or value function is consistent with the true RL solution. When the true distribution is not in the model class, $B_{\phi_\omega^\star}$ converges to the closest representation of the true Bellman operator according to the parametrisation that maximises the likelihood $\mathbb{E}_{P_B(b,s,a|\omega)}\left[\log p(b,s,a|\phi)\right]$. This is analogous to frequentist convergent TD learning where the function approximator converges to a parametrisation that minimises the projection of the Bellman operator into the model class [70, 71, 12]. We now make this relationship precise by considering a Gaussian model.

### 3.2 Gaussian BBO

To showcase the power of Theorem 1 and to provide a direct comparison to existing frequentist approaches, we consider the nonlinear Gaussian model $P(b|s,a,\phi) = \mathcal{N}(\hat{B}_\phi(s,a), \sigma^2)$ that is commonly used for Bayesian regression [55, 33]. The mean is a nonlinear function approximator that best represents the Bellman operator $B_\phi \approx \mathcal{B}[\hat{Q}_\omega]$ and the model variance $\sigma^2 > 0$ represents the aleatoric uncertainty in our samples. Ignoring the log-normalisation constant $c_{\text{norm}}$, the log-posterior is an empirical mean squared error between the empirical Bellman samples and the model mean $\hat{B}_\phi(s_i, a_i)$ with additional regularisation due to the prior (see Appendix D.2 for a derivation):

$$-\log p(\phi|\mathcal{D}_\omega^N) = c_{\text{norm}} + \sum_{i=1}^{N} \frac{(b_i - \hat{B}_\phi(s_i, a_i))^2}{2\sigma^2} + R(\phi), \quad \phi_\omega^\star \in \underset{\phi \in \Phi}{\arg\min} \|\hat{B}_\phi - \mathcal{B}[\hat{Q}_\omega]\|_{\rho,\pi}^2. \quad (9)$$

Theorem 1 proves that in the limit $N \to \infty$, the effect of the prior diminishes and the Bayesian Bellman operator converges to the parametrisation: $\mathcal{B}^\star_{\omega,N} \xrightarrow{a.s.} \hat{B}_{\phi^\star_\omega}$. As $\phi^\star_\omega$ is the set of parameters that minimise the mean squared error between the true Bellman operator and the approximator, $\hat{B}_{\phi^\star_\omega}$ is a *projection* of the true Bellman operator onto the space of functions represented by $\hat{B}_\phi$:

$$\hat{B}_{\phi^\star_\omega} = \mathcal{P}_{\hat{B}_\phi} \circ \mathcal{B}[\hat{Q}_\omega] \coloneqq \{\hat{B}_{\phi'} : \phi' \in \underset{\phi \in \Phi}{\arg\min} \|\hat{B}_\phi - \mathcal{B}[\hat{Q}_\omega]\|^2_{\rho,\pi}\}. \tag{10}$$

Finally, Theorem 1 proves that the MSBBE converges to the mean squared projected Bellman error $\text{MSBBE}_N(\omega) \xrightarrow{a.s.} \text{MSPBE}(\omega) \coloneqq \|\hat{Q}_\omega - \mathcal{P}_{\hat{B}_\phi} \circ \mathcal{B}[\hat{Q}_\omega]\|^2_{\rho,\pi}$. By the definition of the projection operator in Eq. (10), a solution $\hat{Q}_\omega = \mathcal{P}_{\hat{B}_\phi} \circ \mathcal{B}[\hat{Q}_\omega]$ is a TD fixed point; hence any asymptotic MSBBE minimiser parametrises a TD fixed point should it exist. To further highlight the relationship between BBO and convergent TD algorithms that minimise the mean squared projected Bellman operator, we explore the linear Gaussian regression model as a case study in Appendix E, allowing us to derive a regularised Bayesian TDC/GTD2 algorithm [71, 70].

## 4 Approximate BBO

We have demonstrated in Eq. (7) that if it is tractable to sample from the posterior, a simple convergent stochastic gradient descent algorithm can be used to minimise the MSBBE. We derive the gradient update for the linear Gaussian model as part of our case study in Appendix E. Unfortunately, models like linear Gaussians that have analytic posteriors are often too simple to accurately represent the Bellman operator for domains of practical interest in RL. We now extend our analysis to include approximate inference approaches.

### 4.1 Approximate Inference

To allow for more expressive nonlinear function approximators, for which the posterior normalisation is intractable, we introduce a tractable posterior approximation: $q(\phi|\mathcal{D}^N_\omega) \approx P(\phi|\mathcal{D}^N_\omega)$. In this paper, we use randomised priors (RP) [57] for approximate inference. Randomised priors (PR) inject noise into the maximum a posteriori (MAP) estimate via a noise variable $\epsilon \in \mathcal{E}$ with distribution $P_E(\epsilon)$ where the density $p_E(\epsilon)$ has the same form as the prior. We provide a full exposition of RP for BBO in Appendix F, including derivations of our objectives. RP in practice uses ensembling: $L$ prior randomisations $\mathcal{E}_L \coloneqq \{\epsilon_l\}_{l=1:L}$ are first drawn from $P_E$. To use RP for BBO, we write the $Q$-function approximator as an ensemble of $L$ parameters $\Omega_L \coloneqq \{\omega_l\}_{l=1:L}$ where $\hat{Q}_\omega = \frac{1}{L} \sum_{l=1}^L \hat{Q}_{\omega_l}$ and require an assumption about the prior and the function spaces used for approximators:

**Assumption 4** (RP Function Spaces)**.** *i)* $\hat{Q}_{\omega_l}$ *and* $\hat{B}_{\omega_l}$ *share a function space, that is* $\hat{Q}_{\omega'_l} = \hat{B}_{\omega'_l}$ *for any* $\omega'_l \in \Omega$*, where* $\Phi = \Omega \subset \mathbb{R}^n$ *is compact, convex with a smooth boundary. ii)* $\mathcal{E} \subseteq \mathbb{R}^n$ *and* $R(\phi - \epsilon)$ *is defined for any* $\phi \in \Phi, \epsilon \in \mathcal{E}$*.*

For each $l \in \{1 : L\}$, a set of solutions to the prior-randomised MAP objective are found:

$$\psi^\star_l(\omega_l) \in \underset{\phi \in \Phi}{\arg\min} \, \mathcal{L}(\phi; \mathcal{D}^N_{\omega_l}, \epsilon_l) \coloneqq \underset{\phi \in \Phi}{\arg\min} \frac{1}{N} \left( R(\phi - \epsilon_l) - \sum_{i=1}^N \log p(b_i|s_i, a_i, \phi) \right). \tag{11}$$

The RP solution $\psi^\star_l(\omega_l)$ has dependence on $\omega_l$ that mirrors the BBO posterior's dependence on $\omega$. To construct the RP approximate posterior $q(\phi|\mathcal{D}^N_\omega)$, we average the set of perturbed MAP estimates over all ensembles: $q(\phi|\mathcal{D}^N_\omega) \coloneqq \frac{1}{L} \sum_{l=1}^L \delta(\phi \in \psi^\star_l(\omega_l))$. To sample from the RP posterior $\phi \sim q(\cdot|\mathcal{D}^N_\omega)$, we sample an ensemble uniformly $l \sim \text{Unif}(\{1 : L\})$ and set $\phi = \psi^\star_l(\omega_l)$. Although BBO is compatible with any approximate inference technique, we justify our choice of RP by proving that it preserves the consistency results developed in Theorem 1:

**Corollary 1.1.** *Under Assumptions 1-4, results i)-iii) of Theorem 1 hold with* $P(\phi|\mathcal{D}^N_\omega)$ *replaced by the RP approximate posterior* $q(\phi|\mathcal{D}^N_\omega)$ *both with or without ensembling.*

In answer to Question 3), Corollary 1.1 shows that the difference between using the RP approximate posterior and the true posterior lies in their characterisation of uncertainty and not their asymptotic behaviour. Existing work shows that RP uncertainty estimates are conservative [59, 21] with strong empirical performance in RL [57, 58] for the Gaussian model that we study in this paper.

The RP approximate posterior $q(\phi|\mathcal{D}^N_\omega)$ depends on the ensemble of $Q$-function approximators $\hat{Q}_{\omega_l}$ and like in Section 3 we must learn an ensemble of optimal parametrisations $\omega^\star_l$. We substitute

for $q(\phi|\mathcal{D}_\omega^N)$ in place of the true posterior in Eqs. (5) and (6) to derive an ensembled RP MSBBE: $\text{MSBBE}_{\text{RP}}(\omega_l) := \|\hat{Q}_{\omega_l} - \hat{B}_{\psi_l^\star(\omega_l)}\|_{\rho,\pi}^2$. When a fixed point $\hat{Q}_{\omega_l} = \hat{B}_{\psi_l^\star(\omega_l)}$ exists, minimising $\text{MSBBE}_{\text{RP}}(\omega_l)$ is equivalent to finding $\omega_l^\star$ such that $\psi_l^\star(\omega_l^\star) = \omega_l^\star$. To learn $\omega_l^\star$ we can instead minimise the simpler parameter objective $\omega_l^\star \in \arg\min_{\omega_l \in \Omega} \mathcal{U}(\omega_l; \psi_l^\star)$:

$$\mathcal{U}(\omega_l; \psi_l^\star) := \|\omega_l - \psi_l^\star(\omega_l)\|_2^2 \quad \text{such that} \quad \psi_l^\star(\omega_l) \in \arg\min_{\phi \in \Phi} \mathcal{L}(\phi; \mathcal{D}_{\omega_l}^N, \epsilon_l), \tag{12}$$

which has the advantage that deterministic gradient updates can be obtained. $\mathcal{U}(\omega_l; \psi_l^\star)$ can still provide an alternative auxilliary objective when a fixed point does not exist as the convergence of algorithms minimising Eq. (12) does not depend on its existence and has the same solution as minimising $\text{MSBBE}_{\text{RP}}(\omega_l)$ for sufficiently smooth $B_\phi$. Solving the bi-level optimisation problem in Eq. (12) is NP-hard [8]. To tackle this problem, we introduce an ensemble of parameters $\Psi_L := \{\psi_l\}_{1:L}$ to track $\psi_l^\star(\omega_l)$ and propose a two-timescale gradient update for each $l \in \{1 : L\}$ on the objectives in Eq. (12) with per-step complexity of $\mathcal{O}(n)$:

$$\psi_l \leftarrow \mathcal{P}_\Omega\left(\psi_l - \alpha_k \nabla_{\psi_l}\left(R(\psi_l - \epsilon_l) - \log p(b_i|s_i, a_i, \psi_l)\right)\right), \quad \text{(fast)} \tag{13}$$

$$\omega_l \leftarrow \mathcal{P}_\Omega(\omega_l - \beta_k(\omega_l - \psi_l)), \quad \text{(slow)} \tag{14}$$

where $\alpha_k$ and $\beta_k$ are asymptotically faster and slower stepsizes respectively and $\mathcal{P}_\Omega(\cdot) := \arg\min_{\omega \in \Omega}\|\cdot - \omega\|_2^2$ is a projection operator that projects its argument back into $\Omega$ if necessary. From a Bayesian perspective, we are concerned with characterising the uncertainty after a *finite* number of samples $N < \infty$ and hence $(b_i, s_i, a_i)$ should be drawn uniformly from the dataset $\mathcal{D}_{\omega_l}^N$ to form estimates of the summation in Eq. (11), which becomes intractable with large $N$. When compared to existing RL algorithms, sampling from $\mathcal{D}_{\omega_l}^N$ is analogous to sampling from a replay buffer [54]. A frequentist analysis of our updates is also possible by considering samples that are drawn online from the underlying data generating distribution $(b_i, s_i, a_i) \sim P_B$ in the limit $N \to \infty$. We discuss this frequentist interpretation further in Appendix C.5.

To answer Question 4), we prove convergence of updates (13) and (14) using a straightforward application of two-timescale stochastic approximation [15, 14, 42] to BBO. Intuitively, two timescale analysis proves that the faster timescale update (13) converges to an element in $\Omega$ using standard martingale arguments, viewing the parameter $\omega_l$ as quasi-static as it behaves like a constant. Since the perturbations are relatively small, the separation of timescales then ensures that $\psi_l$ tracks $\psi_l^\star(\omega_l)$ whenever $\omega_l$ is updated in the slower timescale update (14), viewing the parameter $\psi_l$ as quasi-equilibrated [14]. We introduce the standard two-timescale regularity assumptions and derive the limiting ODEs of updates (13) and (14) in Appendix C.3:

**Assumption 5** (Two-timescale Regularity). *i)* $\nabla_{\psi_l}R(\psi_l - \epsilon_l)$ *and* $\nabla_{\psi_l}\log p(b_i|s_i, a_i, \psi_l)$ *are Lipschitz in* $\psi_l$ *and* $(b_i, s_i, a_i) \sim \text{Unif}(\mathcal{D}_{\omega_l}^N)$; *ii)* $\psi^\circledast(\omega_l)$ *and* $\omega_l^\circledast$ *are local aysmptotically stable attractors of the limiting ODEs of updates (13) and (14) respectively and* $\psi_l^\circledast(\omega_l)$ *is Lipschitz in* $\omega_l$; *and iii) The stepsizes satisfy:* $\lim_{k \to \infty} \frac{\beta_k}{\alpha_k} = 0$, $\sum_{k=1}^\infty \alpha_k = \sum_{k=1}^\infty \beta_k = \infty$, $\sum_{k=1}^\infty (\alpha_k^2 + \beta_k^2) < \infty$.

**Theorem 2.** *If Assumptions 1 to 5 hold,* $\psi_l$ *and* $\omega_l$ *converge to* $\psi_l^\circledast(\omega_l^\circledast)$ *and* $\omega_l^\circledast$ *almost surely.*

As $\omega_l$ are updated on a slower timescale, they lag the parameters $\psi_l$. When deriving a Bayesian actor-critic algorithm in Section 4.2, we demonstrate that these parameters share a similar role to a *lagged critic*. There is no Bayesian explanation for these parameters under existing approaches: when applying approximate inference to $P(Q^\pi|s, a, \mathcal{D}^N)$, the RP solution $\psi_l^\star$ has no dependence on $\omega_l$. Hence, minimising $\mathcal{U}(\omega_l; \psi_l^\star)$ and the approximate MSBBE has an exact solution by setting $\omega_l^\star = \psi_l^\star$. In this case, $\hat{Q}_{\omega_l^\star} = \hat{B}_{\psi_l^\star}$ meaning that existing approaches do not distinguish between the $Q$-function and Bellman operator approximators.

## 4.2 Bayesian Bellman Actor-Critic

BootDQN+Prior [57, 58] is a state-of-the-art Bayesian model-free algorithm with Thompson sampling [74] where, in principle, an optimal $Q$-function is drawn from a posterior over optimal $Q$-functions at the start of each episode. As BootDQN+Prior requires bootstrapping, it actually draws a sample from the Gaussian BBO posterior introduced in Section 3.2 using RP approximate inference with the empirical Bellman

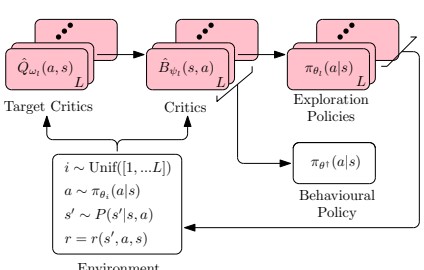

Figure 2: Schematic of RP-BBAC.

function $b_\omega(s',a,s) = r(s',a,s) + \gamma \max_{a'} \hat{Q}_\omega(s',a')$.

This empirical Bellman function results from substituting an optimal policy $\pi(a|s) = \delta(a \in \arg\max_{a'} \hat{Q}_\omega(s,a'))$ in Eq. (3). A variable $l$ is drawn uniformly and the optimal exploration policy $\pi_l^\star(a|s) = \delta(a \in \arg\max_{a'} B_{\phi_l}(s,a'))$ is followed. BootDQN+Prior achieves what Osband et al. [58] call *deep exploration* where exploration not only considers immediate information gain but also the consequences of an exploratory action on future learning. Due its use of the $\arg\max$ operator, BootDQN+Prior is not appropriate for continuous action or large discrete action domains as a nonlinear optimisation problem must be solved every time an action is sampled. We instead develop a randomised priors Bayesian Bellman actor-critic (RP-BBAC) to extend BootDQN+Prior to continuous domains. A schematic of RP-BBAC is shown in Fig. 2 which summarises Algorithm 1. Additional details are in Appendix G.

**Comparison to existing actor-critics:**    Using a Gaussian model also allows a direct comparison to frequentist actor-critic algorithms [50]: as shown in Fig. 2, every ensemble $l \in \{1...L\}$ has its own *exploratory actor* $\pi_{\theta_l}$, *critic* $B_{\psi_l}$ and *target critic* $\hat{Q}_{\omega_l}$. In BBAC, each critic is the solution to its unique $\epsilon_l$-randomised empirical MSBBE objective from Eq. (12): $\mathcal{L}_{\text{critic}}(\psi_l) := -\frac{1}{\sigma^2} \sum_{i=1}^N (b_i - \hat{B}_{\psi_l}(s_i,a_i))^2 + R(\psi_l - \epsilon_l)$. The target critic parameters $\omega_l$ for each Bellman sample $b_i = r_i + \gamma \hat{Q}_{\omega_l}(s_i',a_i')$ are updated on a slower timescale to the critic parameters, which mimics the updating of target critic parameters after a regular interval in frequentist approaches [54, 39]. We introduce an ensemble of parametric exploration policies $\pi_{\theta_l}(a|s)$ parametrised by a set of parameters $\Theta_L := \{\theta_l\}_{l=1:L}$. Each optimal exploration policy $\pi_l^\star(a|s)$ is parametrised by the solution to its own optimisation problem: $\theta_l^\star \in \arg\max_{\theta_l \in \Theta} \mathbb{E}_{\rho(s)\pi_{\theta_l}(a|s)}[B_{\phi_l}(s,a')]$. Unlike frequentist approaches, an exploratory actor is selected at the start of each episode in accordance with our current uncertainty in the MDP characterised by the approximate RP posterior.

Exploration is thus both deep and adaptive as actions from an exploration policy are directed towards minimising epistemic uncertainty in the MDP and the posterior variance reduces in accordance with Corollary 1.1 as more data is collected. BBAC's explicit specification of lagged target critics is unique to BBO and, as discussed in Section 4.1, corrects the theoretical issues raised by applying bootstrapping to existing model-free Bayesian RL theory, which does not account for the posterior's dependence on $\hat{Q}_\omega$. Finally, exploration policies may not perform well at test time, so we learn a behaviour policy $\pi_{\theta^\dagger}(a|s)$ parametrised by $\theta^\dagger \in \Theta$ from the data collected by our exploration policies using the ensemble of critics: $\{\hat{B}_{\psi_l}\}_{l=1:L}$. Theoretically, this is the optimal policy for the Bayesian estimate of the true MDP by using

---

**Algorithm 1** RP-BBAC

Initialise $\Theta_L, \Omega_L, \Psi_L, \mathcal{E}_L, \theta^\dagger$ and $\mathcal{D} \leftarrow \varnothing$
Sample initial state $s \sim P_0$
**while not** converged **do**
    Sample policy $\theta_l \sim \text{Unif}(\Theta_L)$
    **for** $n \in \{1, ...N_{\text{env}}\}$ **do**
        Sample action $a \sim \pi_{\theta_l}(\cdot|s)$
        Observe next state $s' \sim P(\cdot|s,a)$
        Observe reward $r = r(s',a,s)$
        $\mathcal{D} \leftarrow \mathcal{D} \cup \{s,a,r,s'\}$
    **end for**
    $\Theta_L, \Omega_L, \Psi_L \leftarrow$ UPDATEPOSTERIOR
    $\theta^\dagger \leftarrow$ UPDATEBEHAVIOURALPOLICY
**end while**

---

the approximate posterior to marginalise over the ensemble of Bellman operators. We augment our behaviour policy objective with entropy regularisation, allowing us to combine the exploratory benefits of Thompson sampling with the faster convergence rates and algorithmic stability of regularised RL [77].

## 5    Related Work

Existing model-free Bayesian RL approaches assume either a parametric Gaussian [34, 57, 32, 52, 58, 75] or Gaussian process regression model [28, 29]. Value-based approaches use the empirical Bellman function $b_\omega(s',a,s) = r(s',a,s) + \gamma \max_{a'} \hat{Q}_\omega(s',a')$ whereas actor-critic approaches use the empirical Bellman function $b_\omega(s',a',s,a) = r(s',a,s) + \gamma \hat{Q}_\omega(s',a')$. In answering Questions 1-4, we have shown existing methods that use bootstrapping inadvertently approximate the posterior predictive over $Q$-functions with the BBO posterior predictive $P(Q^\pi|s,a,\mathcal{D}^N) \approx P(b|s,a,\mathcal{D}_\omega^N)$. These methods minimise an approximation of the MSBBE where the Bayesian Bellman operator is treated as a supervised target, ignoring its dependence on $\omega$: gradient descent approaches drop gradient terms and fitted approaches iteratively regress the $Q$-function approximator onto the Bayesian Bellman operator $\hat{Q}_{\omega_{k+1}} \leftarrow \mathcal{B}_{\omega_k, N}^\star$. In both cases, the updates may not be a contraction mapping for the same reasons as in non-Bayesian TD [76] and so it is not possible to prove general convergence.

The additional Bayesian regularisation introduced from the prior can lead to convergence, but only in specific and restrictive cases [4, 5, 31, 17].

Approximate inference presents an additional problem for existing approaches: many existing methods naïvely apply approximate inference to the Bellman error, treating $\mathcal{B}[Q^\pi](s,a)$ and $Q^\pi(s,a)$ as independent variables [32, 52, 75, 34]. This leads to poor uncertainty estimates as the Bellman error cannot correctly propagate the uncertainty [56, 57]. Osband et al. [58] demonstrate that this can cause uncertainty estimates of $Q^\pi(s,a)$ at some $(s,a)$ to be zero and propose BootDQN+Prior as an alternative to achieve deep exploration. BBO does not suffer this issue as the posterior characterises the uncertainty in the Bellman operator directly. In Section 4.2 we demonstrated that BootDQN+Prior derived from BBO specifies the use of target critics. Despite being essential to performance, there is no Bayesian explanation for target critics under existing model-free Bayesian RL theory, which posits that sampling a critic from $\tilde{P}(Q^\pi|s,a,\mathcal{D}^N)$ is sufficient.

## 6 Experiments

**Convergent Nonlinear Policy Evaluation** To confirm our convergence and consistency results under approximation, we evaluate BBO in several nonlinear policy evaluation experiments that are constructed to present a convergence challenge for TD algorithms.

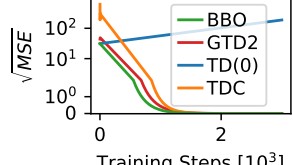

Figure 3: Tsitsiklis counterexample.

We verify the convergence of nonlinear Gaussian BBO in the famous counterexample task of Tsitsiklis and Van Roy [76], in which the TD(0) algorithm is provably divergent. The results are presented in Fig. 3. As expected, TD(0) diverges, while BBO converges to the optimal solution faster than convergent frequentist nonlinear TDC and GTD2 [12]. We also consider three additional policy evaluation tasks commonly used to test convergence of nonlinear TD using neural network function approximators: 20-Link Pendulum [23], Puddle World [16], and Mountain Car [16]. Results are shown in Fig. 11 of Appendix H.3 from which we conclude that i) by ignoring the posterior's dependence on $\omega$, existing model-free Bayesian approaches are less stable and perform poorly in comparison to the gradient based MSBBE minimisation approach in Eq. (7), ii) regularisation from a prior can improve performance of policy evaluation by aiding the optimisation landscape [26], and iii) better solutions in terms of mean squared error can be found using BBO instead of the local linearisation approach of nonlinear TDC/GTD2[12].

**Exploration for Continuous Control** In many benchmark tasks for continuous RL, such as the locomotion tasks from MuJoCo Gym suite [18], the environment reward is shaped to provide a smooth gradient towards a successful task completion and naïve Boltzmann dithering exploration strategies from regularised RL can provide a strong inductive bias.

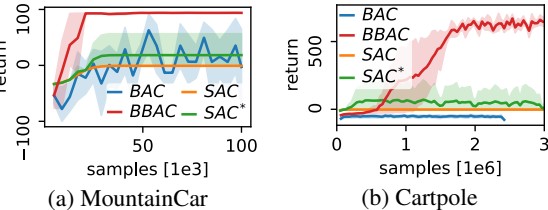

(a) MountainCar     (b) Cartpole

Figure 4: Continuous control with sparse reward.

In practical real-world scenarios, dense rewards are difficult to specify by hand, especially when the task is learned from raw observations like images. Therefore, we consider a set of continuous control tasks with sparse rewards as continuous analogues of the discrete experiments used to test BootDQN+Prior [57]: *MountainCar-Continuous-v0* from Gym benchmark suite and a slightly modified version of the *cartpole-swingup_sparse* from DeepMind Control Suite [73]. Both environments have a sparse reward signal and penalize the agent proportional to the magnitude of executed actions. As the agent is always initialised in the same state, it has to deeply explore costly states in a directed manner for hundreds of steps until it reaches the rewarding region of the state space. We compare RP-BBAC with two variants of the state-of-the-art soft actor-critic: SAC, which is the exact algorithm presented in [40]; and SAC*, a tailored version which uses a single $Q$-function to avoid *pessimistic underexploration* [20] due to the use of the double-minimum-Q trick (see Appendix I for details). To understand the practical implications of our theoretical results, we also compare against BAC which is a variant of RP-BBAC where $\hat{Q}_{\omega_t^\star} = \hat{B}_{\psi_t^\star}$. As we discussed in Section 4.1, BAC is the Bayesian actor-critic that results from applying RP approximate inference to the posterior over $Q$-functions used by existing model-free Bayesian approaches with bootstrapping.

The results are shown in Fig. 4. Due to the lack of smooth signal towards the task completion, SAC consistently fails to solve the tasks and converges to always executing the 0-action due to the action cost term, while SAC* achieves the goal in one out of five seeds. RP-BBAC succeeds for all five seeds in both tasks. To understand why, we provide a state support analysis in for

*MountainCar-Continuous-v0* Appendix I.1, The final plots are shown in Fig. 5 and confirm that the deep, adaptive exploration carried out by RP-BBAC leads agents to systematically explore regions of the state-action space with high uncertainty. The same analysis for SAC and SAC* confirms the inefficiency of the exploration typical of RL as inference: the agent repeatedly explores actions that lead to poor performance and rarely explores beyond its initial state. The state support analysis for BAC in Appendix I.1 confirms that by using the posterior over $Q$-functions with bootstrapping, existing model-free Bayesian RL cannot accurately capture the uncertainty in the MDP. Initially, exploration is similar to RP-BBAC but epistemic uncertainty estimates are unstable and cannot concentrate due to the convergence issues highlighted in this paper, preventing adaptive exploration.

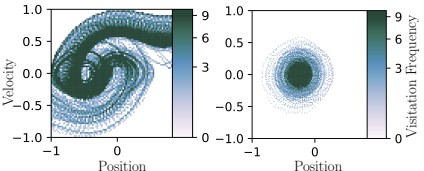

Our results in Fig. 4 demonstrate that the theoretical issues with existing approaches have negative empirical consequences, verifying that it is essential for Bayesian model-free RL algorithms with bootstrapping to sample from the BBO posterior as BAC fails to solve both tasks where sampling from the correct posterior in RP-BBAC succeeds. In Appendix I.2, we also investigate RP-BBAC's sensitivity to randomized prior hyperparameters. The range of working hyperparameters is wide and easy to tune.

Figure 5: State Support for RP-BBAC (left) and SAC (right) in MountainCar-Continuous-v0.

## 7 Conclusion

By introducing the BBO framework, we have addressed a major theoretical issue with model-free Bayesian RL by analysing the posterior that is inferred when bootsrapping is used. Our theoretical results proved consistency with frequentist RL and strong convergence properties, even under posterior approximation. We used BBO to extend BootDQN+Prior to continuous domains. Our experiments in environments where rewards are not hand-crafted to aid exploration demonstrate that sampling from the BBO posterior characterises uncertainty correctly and algorithms derived from BBO can succeed where state-of-the-art algorithms fail catastrophically due to their lack of deep exploration. Future research could experiment with novel inference techniques, more complex priors and likelihoods, especially if they depend on $\omega$, or extend our convergence analysis to the actor-critic algorithm.

## Acknowledgements

This project has received funding from the European Research Council (ERC) under the European Unions Horizon 2020 research and innovation programme (grant agreement number 637713). Matthew Fellows and Kristian Hartikainen are funded by the EPSRC. The experiments were made possible by a generous equipment grant from NVIDIA. We would like to thank Piotr Miłoś, whose proof for a similar problem inspired our proof of Lemma 3.

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
