# OpenReview forum: "Bayesian Bellman Operators"
_NeurIPS.cc/2021/Conference — NeurIPS 2021 Spotlight_

### Official Review · Reviewer_8tDK · 2021-07-05

**Rating:** 8
**Confidence:** 3

**Summary:**

The paper proposes a novel approach to the problem of model-free bayesian reinforcement learning. Interpreting the **update** of the current Q-Function via the Bellman operator as uncertain due to a finite amount of samples, the authors propose to not keep a posterior distribution over Q-Functions but over Bellman operators. This view allows to model the dependency between the posterior over Bellmann operators and the current Q-Function estimate, introducing correction terms to the otherwise ad-hoc bootstrapping common in model free RL. The authors present proofs that the estimated posterior distributions convege to the true Bellman Operator in the limit of infinite data and that gradient-based algorithms for minimizing the mean-squared (bayesian) bellmann error converge.

**Limitations And Societal Impact:**

The Impact has been adequately addressed, especially given that the authors propose an abstract method for reinforcement learning.

**Main Review:**

Strenghts:
- The Bayesian Bellmann operator framework presented in Section 3 is an elegant and theoretically well-motivated view on model-free Bayesian RL. To my knowledge, this view has not been previously proposed in the literature and is a clear and highly important novelty.
- The resulting approximate implementation in Section 4 is also novel and yields further interesting theoretical insights, motivating empirically successful techniques like lagged critics. However, reading up on previous related works (TDC, GTD(0)/GTD(2)) I realized that these interpretations were also existent, although not in a Bayesian setting.
- The presented experiments particularly in sparse-reward, continuous control environments give additional empirical evidence that the proposed method works in practice and further has the promised explorative behaviour.

Weaknesses:
- Unfortunately, the paper is not easy to parse for a reader that is not strongly experienced in both Bayesian RL and theoretical analysis of RL algorithms. Having spent at least some time with Bayesian modelling (although not so much in the RL setting) and quite some time with RL methods (although not their theoretical analysis), I could understand the paper well, although especially parsing details is very hard because it requires a lot of going back and forth between the appendix and the paper. A reason for that may be that the authors have gathered a lot of evidence, maybe even too much to put into a single conference paper. The urge to at least still hint to all that evidence sometimes results in an unsatisfying presentation of certain aspects. To put an example:
    - In Secton 4.2 the authors only spend three lines (314-316) talking about their RP-BBAC algorithm after talking for a much longer time about its connection to BootDQN+Prior. While I personally encourage highlighting such connections, I think that due to the limited space, such connections should have sometimes rather been moved to the appendix in order to allow for more detailed explanations of the proposed approach.
    - I also think that the policy evaluation results in the 20-link pendulum, puddle world and mountaincar environment should have made it into the main paper, given that the Tsitsiklis counter example is clearly important but also more detached from real world applications than the other environments.
- W.r.t to the approach itself, I have only a few points that may turn out as weaknesses:
    - The authors motivate that a Bayesian viewpoint allows to incorporate priors into the method. However, when using complicated approximators such as neural networks, specifying priors in the weight space is fairly unintuitive. So the question is, is the prior in these more complicated architectures really useful from a practical perspective?
    - In Objective (10), the term R(\phi - \epsilon_l) leaves me a bit puzzled. In Section 3, R has been introduced via the prior on the parameters of the posterior approximation, i.e. p(\phi) = exp(-R(\phi)). So what is the meaning of R in objective (10). I assume that this formulation has to do with the use of Randomized Priors but is this rather some ad-hoc choice or does R(\phi - \epsilon_l) in (10) results form re-formulations of the prior in the RP framework?
    - In the continuous control experiments: Why was it necessary to learn the behavioral policy from an average over the Q functions in the ensemble? Did the posteriors and the corresponding Q functions not collapse as nicely as suggested by the theory? Or was there a different reason for learning a policy from the averaged critics?
    - Is it really necessary that \psi_l^*(\omega_l) = \omega_l such that the MSBBE_{RP} is minimized? With a complicated DNN and all its parameter symmetries, shouldn't it be possible to find two values \omega_1 and \omega_2 such that the corresponding Q functions predict the same value? In this case, wouldn't the substitute objective (11) be a more restricted version of objective MSBBE_{RP}?
    - While the authors proof that the approximate inference scheme presented in Section 4 converges, I guess that this does, however, "only" mean that policy evaluation converges? Hence, the RP-BBAC in Section 4.2 is not necessarily guaranteed to converge, although it seems to do in the experiments. Or am I missing something?

Correctness:
- I did not check the individual proofs in the appendix (epsecially not the convergence proofs since this is out of my currernt knowledge of theoretical RL) in all detail. The simpler derivations of e.g. gradients and other formulas in the main paper that I checked looked, however, correct to me.

Clarity:
- What exactly is meant with the statement "\hat{Q}_{w_l} and \hat{B}_{w_l} share a function space" in line 242? Given the following discussion, I guess that it means that the two approximators need to not only be parameterized in the same way but also need to yield the same predictions for the same parameters? For default NNs this would mean that not only the dimensions of the hidden layers need to match but that also the same non-linearities are used? If so, a short clarifying sentence may help. Alternatively, it could also be mentioned in lines 260-262 that this is the reason why the surrogate objective is an interesting idea.
- There are certain formulas which I think are only defined in the appendix, such as \rho(s) in Assumption 1. While readers familiar with RL will probably know how to interpret \rho(s), I think it would be a big help to quickly mention the symbol in lines 139-141, i.e. right before Assumption 1.

Other:
- The equation in line 1147 of the appendix misses a bracket
- In line 234 of the main text, p(\phi \vert D) is written with capital p but it was introduced with lower case p in Eq. 4. Or should capital p indicate an approximation?
- In line 262 of the main text: "minimise" instead of minimising


**Time Spent Reviewing:**

7

---

> ### Author Response · Authors · 2021-08-10
> **Response to Reviewer 8tDK**
>
> We thank the reviewer for their detailed review and insightful comments. We are grateful for them taking the time to carefully review our work. We like their suggestions regarding introducing BootDQN+Prior and replacing the Tsitsiklis counter example with the nonlinear policy evaluation experiments and will carry out these recommendations spaces permitting.
>
> With regards to their questions:
>
> 'Is the prior in these more complicated architectures really useful from a practical perspective?' Practically, we see that as well as encoding our prior beliefs, it also helps to introduce weight regularisation that stabilises our algorithms. When a priori information is given, for example through transfer learning of a value function $Q_\omega^\dagger$ in a similar MDP, if $B_\phi$ and $Q_\omega$ share a function space we can set the prior to the parameters of $Q_\omega^\dagger$. When little information is available a priori, as described in Section G.3 we can set it to our initialisation of the $Q$-function parameters.
>
> 'the term $R(\phi - \epsilon_l)$': As the reviewer has pointed out, this is an artefact of Randomized Priors posterior approximation. As discussed in Section 4.1, the additional variable $\epsilon_l$ is designed to perturb the MAP estimate by injecting noise into the prior. The collection of MAP estimates then defines an approximate posterior. Mathematically, the artefact is a result of the construction of an optimisation problem that is equivalent to finding the exact posterior in the case of a linear Gaussian model, which Osband et al. describe in [58].
>
> 'Why was it necessary to learn the behavioral policy from an average over the Q functions in the ensemble?' As we show in our state-support analysis, the $Q$-functions did collpase as suggested, however, also as expected, this was only after a significant number of samples - if this were not the case it would be concerning that the uncertainty estimate would be too certain too quickly. At test time however, we are interested in exploiting the best Bayesian estimate of the $Q$-function as the agent is not exploring. This means that BBO uses the full ensemble to marginalise over all $Q$-functions rather than using any residual variance in their value for effective exploration. We see this separation means that the BBO agent can learn an optimal policy whilst there is still residual uncertainty in the $Q$-function (i.e. before the $Q$-function ensembles have converged), which further improves data efficiency.
>
> 'Only policy evaluation converges': yes, this is correct. Proving full convergence of the actor-critic is unfortunately beyond the scope of one paper. We'll make this limitation clearer and include it as a direction for future research.
>
> 'What exactly is meant with the statement "\hat{Q}{w_l} and \hat{B}{w_l} share a function space" in line 242?' The reviewer is correct in their interpretation here and we'll clarify this as suggested.

---

### Official Review · Reviewer_NoL3 · 2021-07-14

**Rating:** 8
**Confidence:** 3

**Summary:**

The paper proposes a novel Bayesian treatment of the Bellman operator. The proposed formalism helps overcome significant limitations of previous approaches, where one typically infers a posterior over state-action values or transition functions. Instead, the authors focus on inferring posteriors over the Bellman operators (b = r_s’as + γ Q(s’, a’)). This formulation is attractive as it is free of strong assumptions and is malleable to theoretical analysis. The authors provide concrete (approximate and exact) instantiations of RL methods relying on the Bayesian Bellman operator (BBO), and demonstrate their empirical efficacy.


**Limitations And Societal Impact:**

The limitations and potential negative societal impact is adequately addressed.

**Main Review:**

## Significance

The paper satisfactorily addresses an important problem in Bayesian RL. All formulations I know of that combine bootstrapping with Bayes make strong assumptions that do not hold typically, which is often detrimental to the quality of the resulting algorithms. The formulation proposed by the authors overcomes these problems, is sound, and is supported by theoretical as well as empirical results. This is a very significant contribution.

## Originality

Even though bootstrapping and Bayesian inference have been combined before, it was done so somehow inadequately. This paper proposes a better way of treating bootstrapping in a Bayesian framework. The ideas are novel.

## Quality

The authors are rigorous in evaluating their proposal, especially theoretically. The experimental evidence could be stronger, but this is not a major issue, as the theoretical contributions are strong. The paper is rich in new theoretical results and insights, examples, and implementations.

A concern I have is that $\phi^*_\omega$ can be non-unique, especially with neural networks. How does this affect stability?


## Clarity

The paper is written somewhat clearly, but I had occasional trouble understanding the formalism. The notation gets complicated quite quickly in Section 3, and some variables aren’t introduced very clearly. For instance, $\hat{B}_\phi$ is defined as a conditional expectation over b. But then, what does “The conditional mean of the distribution $\hat{B}_\phi$” (line 134) mean?

It is not always clear what is a random variable and what is a value. For instance, what “$\hat{Q}_\omega$ is known but we are uncertain of its value under the Bellman operator …” means is unclear to me.

Notation is sometimes overridden. Eg, $b_\omega$ is a function but $b_i$ is random variable. This further complicates things.

A table of notation would help the exposition greatly.

**Time Spent Reviewing:**

5

---

> ### Author Response · Authors · 2021-08-10
> **Response to Reviewer NoL3**
>
> We thank the reviewer for their time reviewing our paper and we hope that we can clarify the few concerns that they had:
>
> $\phi^*\_\omega$ can be non-unique, what effect does this have on stability: following [42], our two timescale analysis is designed to handle such situations as we only require the existence of a local, not global, aysmptotically stable attractor of the limiting ODE within the domain of attraction. As discussed in Section B3, this can be ensured by the introduction of a weight decay term (in BBO's case via prior regularisation) which increases the eigenvalues of the Hessian around a fixed point and improves stability of the algorithm. This was confirmed in practice by our policy evaluation experiements which we discuss in line 386 and further in Section G.3.6: we see from Fig. 11 that prior regularisation has a strong effect in stabilising performance of the algorithm. We'll highlight this insight futher in the main section of the paper.
>
> We only refer to $\hat{B}_\phi$ as the conditional mean. It is always defined as $\hat{B}_\phi(s,a):=\mathbb{E}\_{P(b\vert s,a,\phi)}[b]$.
>
> By "$\hat{Q}\_\omega$ is known but we are uncertain of its value under the Bellman operator …” what we mean is that for any $(s,a)$, we can evaluate $\hat{Q}\_\omega(s,a)$ but we don't know a priori what $\mathcal{B}\[ \hat{Q}\](s,a)$ is as we have uncertainty in the transition distribution and reward function. We'll clarify this.
>
> Clarity: We appreciate the reviewer's request for a table of notation. We think that this is a great idea and agree that it would help the exposition.

---

> > ### Comment · Reviewer_NoL3 · 2021-08-30
> > **Reply to the authors**
> >
> > Thank you for the clarifications. I am going to keep my original score (8) as I am already happy with the paper overall.

---

### Official Review · Reviewer_TpMr · 2021-07-16

**Rating:** 7
**Confidence:** 4

**Summary:**

This paper presents a novel approach to model-free Bayesian reinforcement learning by directly modelling the uncertainty over the Bellman operator in a Markov decision process given experience data. The proposed approach accounts for the fact that unbiased estimates of the true Q-function are usually unavailable and instead relies on estimates of a model-dependent Bellman operator. Theoretical results prove the asymptotic consistency of the posterior over such operator, which allows for learning the Bellman operator of the true system via stochastic gradient descent whenever possible to sample directly from this posterior. The paper also provides an implementable algorithm for cases when the true posterior needs to be approximated, which comes equipped with theoretical convergence guarantees. Experimental results complement the theoretical analysis showing that the proposed framework outperforms conventional frequentist baselines in continuous control tasks.

**Limitations And Societal Impact:**

Limitations of the proposed framework were not thoroughly discussed. For instance, there was no discussion on computational complexity of the proposed RP-BBAC algorithm in the estimation of the Bellman operator posteriors, though a per-step complexity of the gradient descent method is briefly commented on (line 268). The plausibility of some of the assumptions (esp. 4 and 5) in practical applications could be discussed. Societal impact is discussed in Appendix A.

**Main Review:**

The paper is well written and provides an interesting solution to drawbacks in existing Bayesian reinforcement learning frameworks. Bayesian RL methods have the potential to provide robust solutions to real-world RL applications by directly taking into account both epistemic and aleatoric uncertainty. The theoretical contributions are clear and provide insights into existing frameworks. Experimental results, though limited, demonstrate the potential of the proposed framework for RL applications. Therefore, I believe this paper should be accepted. However, I elaborate below on a few issues.

1. The proposed approach does not directly model a (prior) distribution over $\omega$, the parameters of the Q-function, relying instead on (stochastic) gradient descent to approximate the true Q-function with a point estimate at an $\omega^*$ minimising a squared error term (MSBBE). Have the authors considered a more complete Bayesian approach where a distribution over $\omega^*$ would be estimated?
2. There are no theoretical results on how far the Q-function formulated with the minimiser of MSBBE is to the true Q-function. I believe the paper would benefit of a discussion on this final Q-function approximation error, which I have missed.
3. The experiments section could benefit from comparisons against other Bayesian RL approaches, such as VariBAD [81], since currently it only presents comparisons against traditional frequentist baselines.
4. Some of the notation is confusing in equations which end up somewhat cluttered, leading to a few possible typos. For instance, in Eq. 1's expectation subscript, should it be $a'$, instead of $a$, in the probability? In line 119, should it be a conditional probability $P^\pi(s',a'|s,a)$? Theorem 1, line 193, apparently has $\phi^*$ missing $\omega$ in the subscript. Lastly, Eq. 9 is quite confusing. It'd be better to use another notation for the argmin solution, not the same as the argument $\phi$. Also, $\hat{B}_\phi$ is the same notation inside the set and in the subscript of $\mathcal{P}_{\hat{B}_\phi}$.
5. Ref. 22 is missing a full authors list.

**Time Spent Reviewing:**

3

---

> ### Author Response · Authors · 2021-08-10
> **Response to Reviewer TpMr**
>
> We thank the reviewer for their considered and detailed feedback, as well as their insights into our work.
>
> 1. This is an interesting idea, however from what I believe you are suggesting, estimating a distribution over $\hat{Q}_\omega$ would mean introducing a distribution over Q-functions, which would somewhat defeat the point of the framework as this is not practical for the reasons outlined in Section 2.3, and indeed is what we are avoiding in BBO. One idea that we have had would be to introduce eligibility traces into our framework which would allow us to toggle how much we rely on the pushforward through $\hat{Q}_\omega$, however we believe developing this lies far beyond the scope of a single paper.
>
> 2. 'There are no theoretical results on how far the Q-function formulated with the minimiser of MSBBE is to the true Q-function' In Theorem 1, we proved that the Bayesian Bellman operator converges to $B\_{\phi^\star_\omega}$. $\phi^\star_\omega$ is well defined as the minimiser of the KL divergence between the true pushforward $P_B(b\vert s,a,\omega)$ and the model $P(b\vert s,a,\phi)$. Under the Gaussian model defined in Section 3.2, we proved that the minimiser to the MSBBE asymptotically corresponds to the minimiser of the MSPBE used in frequentist RL, whose relationship to the true $Q$-function has been well characterised [22].  We'll make this relationship clearer. In the more general setting, it is obviously not possible to analyse this solution without making assumptions on the model $P(b\vert s,a,\phi)$. In this sense, BBO opens the door for future work to minimise new families of errors that corrrespond to families of models $P(b\vert s,a,\phi)$. We look forward to pursuing this line of research.
>
> 3. Whilst a comparison to VariBAD is certainly interesting, we note that VariBAD is a model-based Bayesian RL algorithm. For this paper, the purpose of our experimental section was to verify our theory which analyses model-free RL, rather than compare model-free to model-based Bayesian RL. In terms of comparison to other Bayesian methods, our framework is general enough that we can derive the BBO analogue of model-free Bayesian approaches simply by choosing the appropriate likelihood, prior and method of approximate inference. We thus compared to BAC, which is a (naive!) Bayesian method that does not account for bootstrapping, but does characterise uncertainty in a $Q$-function using bootstrapped samples. This validates the necessity for using the BBO posterior instead of a posterior over $Q$-functions.
>
> 4. Thank you for spotting and correcting these typos, you are correct in your interpretation and we will change them. We will correct the notation of $\hat{B}_\phi$ in the set.
>
> 5. Thank you for spotting this, we will correct it.
>
> Other comments: we discuss the plausibility of all assumptions in Appendices B1 and B3. In particular, we discuss how Assumption 4 is satisfied trivially if the parameter space is a Euclidean ball, which is simple to enforce for the majority of function approximators, and we provide a standard discussion of Assumption 5.

---

> > ### Comment · Reviewer_TpMr · 2021-08-31
> > **Reply to clarifications**
> >
> > Thanks for the clarification! I am keeping my vote for acceptance.

---

### Official Review · Reviewer_Zz7u · 2021-07-20

**Rating:** 7
**Confidence:** 3

**Summary:**

The authors introduce a Bayesian perspective on the application of Bellman operators to estimated Q-functions. The posterior distribution over the operator application is described, a Bernstein-von Mises-style frequentist analysis is undertaken, a variety of approximate approaches are proposed, and the methods are evaluated on a variety of prediction & control tasks.

**Limitations And Societal Impact:**

As mentioned above, further discussion of the limitations of the work would be welcome.

**Main Review:**

Post-rebuttal:

I thank the authors for their detailed responses to the queries I raised. Many of the points I mentioned in the original review were clarified, and in my opinion the paper contributes several interesting ideas and algorithmic developments, and I am happy to recommend acceptance for the paper.

---

The paper is generally written clearly (although is somewhat dense). Bayesian approaches to reinforcement learning have theoretically interesting properties but without approximation are computationally extremely costly, so the general topic of the paper is likely to be of interest to the RL community at NeurIPS.

This is a dense paper with several distinct contributions. I liked the discussion around distinguishing dependence on bootstrapping in calculating the posterior distribution for the parameters phi and downstream estimators. I felt some sections of the paper were too terse, such as Section 4.2, where the details of the proposed algorithm, RP-BBAC, are given almost entirely in the appendix. There is also an extensive set of experiments carried out (and primarily reported in the appendices). While some of the experimental results in the main paper potentially leave questions open as to why BBO provides stability in comparison to TD (i.e. is it a case of regularization rather than uncertainty estimates?), the authors do provide further analysis (e.g. in Section H.1) that delves into this question further and does seem to support the hypothesis that uncertainty quantification is important.

I have a few questions about some of the contributions of the paper - these are the primary reason for not giving a higher rating, and I would be happy to revise my rating once the authors have responded to these:
 * The sentence beginning after Eqn (4) seems crucial. At this point, I felt the narrative was a little unclear as to what predictions the posterior distribution is to be used for. A natural estimate in Bayesian RL might be of Q^*, but the authors focus on T^pi Q_\omega - presumably this is much easier computationally, but I wasn't sure what the end-goal of this prediction would be at this stage. Similarly, I wasn't sure why there was a focus on the posterior mean at this point. The posterior mean is often motivated as the Bayes estimator for squared loss in non-sequential settings, but I wasn't sure what the motivation is for its consideration here.
 * I didn't see much discussion of how the prior should be set in the paper. This seems to be an important issue, since unlike Bayesian approaches that specify e.g. a single prior over the optimal Q-function, here the prior depends on the function approximation parameters omega, so it is potentially a more difficult decision to decide how to set it.
 * Can the authors clarify what aspects of the analysis leading to Theorem 1 is novel beyond standard applications of Bayesian consistency analysis? Also further comments on the strength of the assumptions/applicability of the results would be useful here - assuming a density for p(b | s , a, phi) seems somewhat strong, and I think rules out the environments considered in the experiments.
 * Experimental details: I found some of the experiments in the main paper to be missing, even when consulting the appendix. For example, I couldn't details of the prior (or the assumed likelihood hyperparameters) for the Tsitsiklis example in the main paper or the appendix.

Minor comments:
 * Line 200: This claim (that derived properties such as optimal policies are also consistent) requires some more justification/qualification (i.e. that the function of the Bellman operator under consideration is continuous, for example); the claim does hold for value functions, but even for optimal policies requires some care.
 * In Figure 3, BBO seems to start with better MSE than GTD2/TDC; is the improved performance explainable due to better initialization?

**Time Spent Reviewing:**

4

---

> ### Author Response · Authors · 2021-08-10
> **Response to Reviewer Zz7u**
>
> Thank you for your time reviewing our work, we appreciate the effort taken. Please see responses to your questions below:
>
> Q1: The BBO predictive. To clear up confusion here, maybe it is worth reiterating the purpose of BBO: as stated in Section 2.3, the ideal situation in model-free Bayesian RL, and what is assumed by all current research, would be to obtain samples of a noisy $Q$-function. If we had some policy $\pi$ for example, we could conceive of a method to do this by letting the agent interact with the environment and sampling the resulting discounted return starting from a state-action pair $(s_i,a_i)$: $q_i=r_i+\gamma r_{i+1} +\gamma^2 r_{i+2}+...$. Each sample $q_i$ is then the true $Q$-function $Q^\pi(s_i,a_i)$ at $(s_i,a_i)$ corrupted by noise $q_i=Q^\pi(s_i,a_i) + \eta_i$ where the distribution over the noise is the pushforward of the distribution over trajectories starting from $(s_i,a_i)$ through the return. We could then use these samples to infer a posterior over $Q^\pi$. Clearly this is not feasible for the same reasons in frequentist RL: requiring unbiased samples of the $Q$-function via the return is impractical and the variance in our samples will likely be very high. This is made even more unrealistic if we are trying to obtain samples from $Q^*$ as we have no way of sampling from an optimal policy a priori. To be of practical use, Bayesian methods typically introduce some degree of bootstrapping, replacing each sample $q_i$ with an approximation e.g. $q_i\approx r_i+\hat{Q}\_\omega(s_{i+1},a_{i+1})$ or  $q_i\approx r_i+\max\_{a'\in\mathcal{A}} \hat{Q}\_\omega(s_{i+1},a')$ if we are trying to approximate $Q^*$. However in Bayesian approaches, this is particularly problematic as samples are no longer noisy estimates of a $Q$-function and so the posterior inferred will no longer be a predictive over a $Q$-function. This presents the four key theoretical questions that need to be addressed if we are to use model-free Bayesian RL and that motivate our research as outlined in the list on line 95: 1. Do model-free Bayesian RL approaches that use bootstrapping still infer a posterior?, 2. If it exists, how does this posterior relate to solving the RL problem?, 3. What effect does approximate inference have on the solution?, 4. Do methods that sample from an approximate posterior converge?
>
> So it is not our choice to 'focus on $\mathcal{B}[Q_\omega]$' (we've used our notation here for consistency). Rather our analysis has revealed that the true posterior that we are actually inferring (and crucially what existing methods are inferring) when using bootstrapping with model-free Bayesian RL is a posterior over $\mathcal{B}[Q_\omega]$.
>
> As to why we introduce the posterior (predictive) mean, observe that the posterior $P(\phi\vert \mathcal{D}_\omega^N)$ depends on $\omega$ as our samples depend on $\omega$ because we have explicitly accounted for bootstrapping. But what value should $\omega$ take? To resolve this, and as we state on 153, 'we must therefore introduce a method of learning the correct $Q$-function approximator'. Existing methods don't explictly face this issue as they assume it is possible to infer a posterior over $Q^\pi$ or $Q^*$. A natural Bayesian estimate of the true value of the Bellman operator $\mathcal{B}[Q_\omega]$ is the predictive mean, which we refer to as the Bayesian Bellman operator, and so we require that the $Q$-function approximator satisfies a (Bayesian) Bellman equation as described on lines 157-159. We're not sure what you mean by: 'posterior mean is often motivated as the Bayes estimator for squared loss'. We can understand that in the specific case of a Gaussian model, the predictive mean will correspond to the solution to a (regularised) mean squared error (and indeed in Section 3.2 we prove this is true) but we don't assume a Gaussian model in BBO. As stated in the paper (line 155) the predictive mean is the correct choice for deeper reasons: the predictive mean accounts for our uncertainty in the true Bellman operator by marginalising over all possible paramterisations of Bellman operators.
>
> Q2. The Prior: It is certainly possible that the prior could depend on $\omega$, but actually what that prior is encoding is our knowledge of the true Bellman operator $\mathcal{B}[\hat{Q}\_{\omega^\star}]$ where $\omega^\star$ minimises the MSBBE in the limit $N\rightarrow\infty$: $\omega^\star \in \arg\_{\omega\in\Omega}\min \lVert \hat{Q}\_\omega -\hat{B}\_{\phi^*\_\omega}\rVert$ (we've used the result from Theorem 1 here to find the limit). This reflects the requirement that the $Q$-function approximator should satisfy a Bayesian Bellman equation. To understand what this translates into practically, consider the example of Gaussian BBO. As we show in Section 3.2, $\mathcal{B}[\hat{Q}_{\omega^\star}]$ is a TD fixed point, hence the Gaussian BBO prior is similar to having single prior over the Q-function in the classic model-free Bayesian RL formulation except now it represents our knowledge of the TD fixed point instead of the true Q-function. Like in classic model-free Bayesian RL formulation, in cases wehere there is little a priori knowledge about this solution, we expect the prior to be either uninformative, or have a convenient form such as a Gaussian with a large variance. In cases where transfer learning is taking place for example, a good prior may be a Gaussion with the mean as the learnt $Q$-function approximator in a similar domain. Thank you for raising this, it is an interesting point and we will introduce this example in the discussion of Gaussian BBO in Section 3.2.
>
> Q3: Our Analysis in Theorem 1: The reviewer is correct to point out that our analysis is an application of Bayesian consistency analysis (under model misspecification). We acknowledge this on line 185 where we state: 'our main result is a Bernstein-von-Mises-type theorem [48] applied to reinforcement learning.' What makes our application challenging and unique is that 1) we allow our samples to be generated from a Markov chain and 2) our analysis explicitly deals with the dependence of the samples on $\omega$, which allows us to formally draw conclusions about the aysmptotic solution, for example, in our derivation of the TD fixed point in Section 3.2 when using a Gaussian model. Moreover, most papers that carry out consistency analysis under misspecification require the reader to verify a local asymptotic normality (LAN) condition (see the paper cited [48]). This is a technically challenging assumption that may be too abstract to be practical or intuitive in an RL setting and requires a rigorous understanding of advanced probabilistic measure theory to verify. Our proof in BBO requires verifying assumptions that are commonplace in the RL literature.
>
> We disgree that the assumption of a density for $p(b | s , a, \phi)$ is strong: to prove this assumption holds, we only need to show that the Radon-Nikodym property holds for the distribution $P(b | s , a, \phi)$. This is thus very weak assumption and we can't think of a parametric distribution that is of practical use where this does not hold: to see why, let $x$ be a random variable with a known density $p(x)$ (e.g. Gaussian in the continuous case, multinomial in the discrete case...), we can generate arbitrarily complex distributions using the change of variables $y=f(x;\phi)$. In this case, it is well known that the Radon-Nikodym property holds provided $f(x;\phi)$ is measurable under which conditions $p(y\vert \phi)$ will exist. Indeed, if the transformation is monotonic and differentiable (a much stronger assumption!), we can derive the density exactly using the classic change of variables formula and generate densities using e.g. normalising flows. For all of our experiments we assume a Gaussian distribution, so for example for our control experiment we have a density :
>
> $$p(b\vert s,a,\phi)\propto \exp\left( - \frac{1}{2\sigma^2}(b-\hat{B}_\phi(s,a))^2\right) $$
>
>
> Q4: Experimental details. Like our other policy evaluation experiments we assumed a Gaussian prior and likelihood for the Tsitsiklis example. We stated this in line 377 but will clarify the use of a Gaussian prior further and also introduce a table of all likelihoods and priors for all experiements in the Appendix.

---

> > ### Comment · Reviewer_Zz7u · 2021-08-14
> > **Thank you for the clarifications - some remaining questions**
> >
> > I thank the authors for their detailed response to my initial review. The response has helped to clarify a few of the queries raised in the initial review, and I will update my rating for the paper at the end of the discussion period. There are still a few points of uncertainty that remain, as described below.
> >
> > Q1.
> >
> > The authors' response here was very helpful. I think a source of confusion in the submission is that the posterior is parameterized by \omega. To me, a much more natural way to present things would be to have the posterior be over the Bellman operator (which does not depend on \omega at all), take the posterior mean operator (which also does not depend on \omega), and then ask to find Q_\omega that minimizes the MSBE for this posterior mean operator. For me, presenting things in this way would help clarify the paper substantially; I'm not aware of prior work in Bayesian inference that works with parameterized posteriors in this way, but if the authors are, these could be very useful to include as references in the main paper to help readers unfamiliar with this approach.
> >
> > Hopefully this perspective makes my comments about the posterior mean clearer too. For example, the posterior over the Bellman operator gives rise to a posterior over Q* too (since Q* is a function of the operator). However, as far as I understand, working with the posterior mean Bellman operator is not equivalent to working with the posterior mean for Q*, since the relationship between the optimal Q-function and the Bellman operator is non-linear.
> >
> > This is one of the reasons I would be interested in more discussion of why the posterior mean, specifically of the Bellman operator, is the right/a sensible choice. In the authors' response, they state:
> >
> > "A natural Bayesian estimate of the true value of the Bellman operator is the predictive mean."
> >
> > I think this is the key point: some more discussion of why this estimator is natural, and further discussion of why it is preferred (in this paper) to other estimators would be useful. It could be argued that we might prefer to work the posterior mean of Q* rather than the posterior mean of the Bellman operator, for example. This might be computationally difficult, which might then form part of an argument for choosing to work with the posterior mean of the operator instead.
> >
> > To expand on the comment "posterior mean is often motivated as the Bayes estimator for squared loss", a standard way to motivate the posterior mean as an estimator for an unknown parameter in the Bayesian decision-theoretic framework is to specify a squared loss L(\hat{theta}, theta) = \| \hat{\theta} - \theta \|^2, and ask for the estimator \hat{\theta} that minimizes this loss in expectation, where the expectation is over \theta distributed according to the posterior in question. Since the function from Bellman operator to fixed-point is non-linear, it wasn't clear to me why using the posterior mean over the Bellman operator is obviously the right thing to do, as described above. For example, an alternative objective would be to minimize (with respect to omega) the expectation (with respect to the posterior on the Bellman operator) over the MSBE, which I don't think is equivalent to minimizing the MSBE for the posterior mean Bellman operator, but would welcome clarification on this.
> >
> > Q2.
> >
> > Thank you for the clarifications here. I'm still not sure if I understand this point entirely however. At Line 145 of the submission, it's stated that
> >
> > "We represent our preexisting beliefs in the true Bellman operator by specifying a prior P(phi) with a density p(phi) which assigns mass over parametrizations of function approximators in accordance with how well we believe they represent B[\hat{Q}_\omega]."
> >
> > Because of this, I'm not sure whether the prior is over the true Bellman operator (which would be independent of \omega) or over B[\hat{Q}_\omega] (which seems like it should depend on \omega), or over B[\hat{Q}_\omega*], as the authors mention in their response, which would also be independent of \omega, but is not a prior over the Bellman operator, but rather a prior over an approximate fixed point of the operator, which doesn't seem to match the way the posterior mean over the operator is treated in the discussion above. Can the authors clarify which of these is the intended interpretation of the prior?
> >
> > Q3.
> >
> > Thank you for the clarifications here. I think some specific examples where the assumption of a density for p(b|s, a, phi) seems restrictive might include the case of known deterministic rewards, or a spike-and-slab prior on rewards (e.g. with a spike corresponding to 0), although would welcome clarifications from the authors if this is not correct. Having said this, I appreciate that the assumptions of densities are standard for these kinds of results. Is the use of a Gaussian prior compatible with Assumption 2 (specifically the compactness of Phi)? To the authors point around push-forwards through measurable transformations f(x; phi), I think some kind of assumptions on f might be necessary here, since taking f to be a singular function results in a distribution without density.

---

> > > ### Author Response · Authors · 2021-08-17
> > > **Further Clarifications**
> > >
> > > Thank you for your response. We think it might futher clarify your remaining points of uncertainty to reiterate the fact that, like all model-free Bayesian RL approaches, we are carrying out Bayesian regression in BBO. In existing approaches, this is Bayesian regression over the unknown Q-function (we make this point on lines 72 and 77). To account for bootstrapping, in BBO we are carrying out regression over the unknown function $\mathcal{B}[\hat{Q}_\omega]$ as this is what we are restricted to sample from. It is well established in Bayesian regression that the posterior predictive mean represents the model's expected prediction of an output given any input and the predictive variance represents the uncertainty in its value [1][2], hence in BBO, our current best Bayesian estimate of the underlying unknown function $\mathcal{B}[\hat{Q}_\omega]$ is the predicitive mean. We don't believe there is anything controversial about this statement and we can emphasise it on line 154 for clarity. When function approximators are used in classic frequentist RL, a Bellman equation $\mathcal{B}[\hat{Q}_\omega]=\hat{Q}_\omega$ is solved. In BBO, we recognise that we don't have access to $\mathcal{B}[\hat{Q}_\omega]$ because of our uncertainty in the Bellman operator and so it is natural to use the predicitive mean, which is our model's expected prediction of $\mathcal{B}[\hat{Q}_\omega]$, in its place.
> > >
> > > 'To me, a much more natural way to present things would be to have the posterior be over the Bellman operator (which does not depend on \omega at all), take the posterior mean operator (which also does not depend on \omega), and then ask to find Q_\omega that minimizes the MSBE for this posterior mean operator'
> > >
> > > If our interpretation here is correct, we believe that what you are suggesting is essentially solving a planning problem in a BAMDP: to model the uncertainty in the Bellman operator as you suggest, we would need to infer and maintain posteriors over the reward and transition distribution. The posterior mean operator would then be obtained by marginalising out over rewards and transitition distributions using the posteriors. Unfortunately, unless there is something we have missed, your suggestion means we will still be burdened with the intractability of solving a BAMDP due to the marginalisation, modelling of the transition distribution and high dimensional integrals required for planning using the Bellman operator, which is what we are explicitly trying to avoid by taking a model-free approach.
> > >
> > > For the prior, as you point out and as we acknowledged in our response, it can depend on $\omega$ to mirror the fact it represents beliefs over $\mathcal{B}[\hat{Q}_\omega]$. This won't affect our theoretical results as the influence of the prior diminishes as $N\rightarrow \infty$. However, we decided to drop the dependence on $\omega$ for the prior because when solving the RL problem, our goal is to ultimately infer the posterior over $\mathcal{B}[\hat{Q}\_{\omega^*}]$, hence using any knowledge we have about the solution $\mathcal{B}[\hat{Q}\_{\omega^*}]$ in the prior clearly will speed learning (this is something we observed in our experiments) and we want to represent this belief across all $\mathcal{B}[\hat{Q}_\omega]$ regardless of $\omega$. We feel that both perspectives are valid and we are happy to include a discussion about the possibility of the prior depending on $\omega$ as well as clarifying our choice for ignoring this dependency. We think this is an interesting point and thanks for raising it. Indeed, experimenting with more general priors that are functions of $\omega$ is an appealing direction for future research, especially as they could provide regularisation that penalises $\hat{Q}_\omega$ and $\hat{B}_\phi$ differing from each other.
> > >
> > > [1] Gal Y., Uncertainty in Deep Learning, PhD Thesis 2016
> > >
> > > [2] Murphy K., Machine Learning: a Probabilistic Perspective MIT Press, 2012

---

### Decision · Program_Chairs · 2021-09-27

**Decision:**

Accept (Spotlight)

**Comment:**

All of the reviewers agree that the paper presents a novel approach to model-free Bayesian RL that distinguishes itself from previous work with a clear theoretical construction and empirical evidence to support the approach. Given the potential importance and interest to the community, I recommend acceptance as a spotlight.